# From Prescription Drugs to Natural Health Products: Medication Use in Canadian Infants

**DOI:** 10.3390/children9101475

**Published:** 2022-09-27

**Authors:** Pascal Bedard, Geoffrey L. Winsor, Emma S. Garlock, Meghan B. Azad, Allan B. Becker, Piush J. Mandhane, Theo J. Moraes, Malcolm R. Sears, Stuart E. Turvey, Padmaja Subbarao, Fiona S. L. Brinkman, Anita L. Kozyrskyj

**Affiliations:** 1Department of Pharmacy, CHU Sainte-Justine, Montreal, QC H3T 1C5, Canada; 2Department of Molecular Biology and Biochemistry, Simon Fraser University, Burnaby, BC V5A 1S6, Canada; 3Department of Pediatrics and Child Health, University of Manitoba, Winnipeg, MB R3A 1S1, Canada; 4Children’s Hospital Research Institute of Manitoba, Winnipeg, MB R3E 3P4, Canada; 5Department of Pediatrics, University of Alberta, Edmonton, AB T6G 1C9, Canada; 6Department of Pediatrics, Hospital for Sick Children, Toronto, ON M5G 1X8, Canada; 7Department of Medicine, McMaster University, Hamilton, ON L8P 1H6, Canada; 8Department of Pediatrics, BC Children’s Hospital and The University of British Columbia, Vancouver, BC V6H 0B3, Canada

**Keywords:** pediatrics, drugs, medicine and pharmaceutical industry, pharmacy, family medicine, general practice, primary care

## Abstract

Limited data exist on pharmaceutical product use by infants, although available data suggests higher prevalence of use among children under 12 months of age. We conducted a descriptive study of 3050 infants recruited in the CHILD Cohort Study, a prospective, multicenter, longitudinal cohort following children from pregnancy through childhood. Parents were surveyed for use of prescription and over-the-counter drugs, and natural health products (NHPs, including homeopathic products and vitamins) at 3, 6, and 12 months after delivery. By one year of age, 96.0% of children had taken at least one pharmaceutical product. Among 307 reported products, 32 were given to at least 1% of cohort infants. Vitamin D, acetaminophen, ibuprofen, topical hydrocortisone, amoxicillin, and nystatin were the most common medications and natural health products (NHPs) received, with 8/32 of the most frequently used products being NHPs. Overall, 14.7% of pharmaceutical products administered to children were off-label and 35.8% were NHPs or products without a Drug Identification Number (DIN). The use of over-the-counter medications and NHPs is common and off-label use of drugs is frequent, even in the first year of life. This study highlights the importance of conducting studies on medication use in infants, and of infant medication use monitoring by healthcare providers.

## 1. Introduction

There are very few data focusing on medication use in infants, especially outside of the hospital setting. Nevertheless, children aged 0–2 years receive drugs more frequently than all other age groups up to adolescence [1,2,3]. The reported prevalence of medication use in this age group varies widely, ranging from 25–96% [1,4]. There are several reasons why medication use in infants should be considered an important public health issue. First, medication dosing errors can occur if parents make mistakes when administering their children’s medication [5]. Second, many pharmaceutical products do not have sufficient pediatric safety data, especially in the treatment of infants and young children [6,7]. Finally, the risk of experiencing serious consequences from an adverse drug event is approximately three times higher in children than in adults [8]. Judicious use of pharmaceuticals in this age group is therefore required to minimize risks while maximizing anticipated benefits [9,10]. Since an accurate evaluation of medication use in infants and children can tailor health care provider decisions and identify areas requiring additional research, this study aims to describe the use of pharmaceutical products (including off-label) in Canadian infants aged 0–12 months.

## 2. Materials and Methods

This study is based on questionnaires collected within the scope of the CHILD Cohort Study (www.childcohort.ca (accessed on 7 October 2021)), a prospective, general population, multicenter, pregnancy cohort following offspring through childhood [11]. Local Research Ethics Board approval was obtained for the study and pregnant mothers (second trimester) recruited from the general population of 4 Canadian provinces (study centers in the cities of Vancouver, Edmonton, Winnipeg and Toronto) from 2009 to 2012 provided signed informed consent. Maternal sociodemographic information including total family income and the number of years the mother lived in Canada was obtained following recruitment. Infant biological sex, season of birth, and year of birth were obtained from childbirth chart data. When the child was 3–4 months old, research assistants administered a questionnaire on medication from birth to 3 months of age. Parents were asked, “Did your baby take ANY prescribed or over-the-counter medications during this time period?” Data concerning the drug name, dosage, treatment length and reason for use were collected. To facilitate analysis of most common reasons for medication use, responses were standardized using automatic assignment to ontologies which were cross-validated by manual curation taking place between 19 August 2019 and 14 April 2020. The same questionnaire was administered 6 and 12 months after birth to assess medication use during the 4–6-month and 7–12-month periods, respectively. Loss to follow-up was low with a 92% retention rate at 1 year. Age-specific medication usage was based on non-missing survey responses. When identifying infants that had taken at least one medication between 0–12 months, those with missing questionnaire responses were included in the analysis if they had reported taking a medication on at least one of the questionnaires. When assessing how many infants had taken no medications during the same time period, those that explicitly reported taking no medications on each of the 3 questionnaires were included in the analysis. Generic drug and NHPs names were obtained from Health Canada’s Drug Product Database [12], the Licensed Natural Health Products Database [13] or product/company websites. Products were attributed an Anatomical Therapeutic Chemical (ATC) code, using the ATC Index 2015, when possible. [14] Prescription drugs were defined as any drug for which a prescription is needed in Canada, according to the Health Canada website and product monographs. Natural health (including vitamins) and homeopathic products were put in a broad NHP category as is the regulatory practice in Canada. All other products were labeled over the counter (OTC), except for products that were not authorized for use in Canada, which were labeled as foreign products [12]. Vitamin D (cholecalciferol) usage was also obtained from nutrition questionnaires administered concurrently with medication questionnaires which recorded “D-drops or D-Vi-sol (Vitamin D)”, “Poly-Vi-Sol with iron”, “Tri-Vi-Sol (Vitamins A, C, D)” or “Poly-Vi-Sol (Vitamins A, C, D, B1, B2, B3, B6)” use. When parents were unsure of the type of medication taken, questionnaire responses were coded as “Unknown” and no assignment to prescription drug, OTC or NHP groups was performed.

Drug monographs were the source of information for whether medication use in infants was off-label. Off-label use was defined when: (1) drug was given below the minimum age stated in the monograph, (2) product monograph stated a lack of evidence in the pediatric population or (3) drug was contraindicated in children according to the monograph. Natural health and homeopathic products and some other products without a drug identification number (such as moisturizing creams) were considered unlabeled because Health Canada did not require a full monograph at the time of the study. 

Descriptive statistics were used to depict use of pharmaceutical products in different age periods (0–3, 4–6, 7–12 months). A non-parametric Kruskal–Wallis test for differences was performed and followed up by post hoc testing using Dunn’s method. Mutual information (MI) scores were calculated between all pairs of binary variables indicating usage of a specific prescription medication, over-the-counter drug (OTC) or natural health product (NHP) during the first year of life and compared using a complementary Spearman rank correlation approach. MI scores ≥ 0.005 underwent permutation testing (1000 permutations) to assign *p*-values (significance at *p* ≤ 0.05) which were corrected for multiple testing using the Benjamini-Hochberg method. Prescription drugs, OTCs and NHPs associated were grouped into top-level ATC codes, OTC and NHP categories, respectively, and the MI scores were visualized as lines connecting all significant pairs of variables on a chord diagram (or globe) constructed using Circos [15]. 

## 3. Results

From 3264 pregnant women recruited in the General CHILD cohort (Figure 1), 3050 families (93%) answered at least 1 drug-related questionnaire. The 3-month questionnaire had the highest response rate (90%), whereas the 6-month questionnaire had the lowest (61%). Table 1 summarizes the participants’ characteristics which were geographically well distributed between the study centers. 

During 2009–2013, 96% of children were exposed to at least one pharmaceutical product. Children received a pharmaceutical product more often in the 7–12-month period (89.4%) than in the 0–3 month and 4–6-month periods (83.6%, 84.7%, respectively; Table 2). The use of prescription drugs was lowest in 4–6-month time period. A steady increase in OTC product use was observed as the infants grew older, whereas the use of NHPs declined. Children received 2 or more pharmaceutical products most often in the 7–12-month period (Figure 2). Less than 5% of children received more than four products during each time period.

Respondents reported the use of 307 different pharmaceutical products; 32 products were provided to at least 1% of the infants in the cohort (Table 3). Among these, nine were dermatological products (ATC D), eight natural health or homeopathic products (type NHP), five topical anti-infectives, four systemic antibiotics, and four associated with alimentary tract and metabolism (ATC A). The products most used were vitamin D (79%), acetaminophen (67%, most commonly for teething), ibuprofen (20%, teething), hydrocortisone (11%, rash) and amoxicillin (10%, ear infection; Table 3). There were wide variations in use among the three age periods studied. Ibuprofen and amoxicillin use increased 9-fold, and acetaminophen use increased by 25% between the 0–3- and 7–12-month periods. During the same time period, vitamin D use decreased by 24% and simethicone use decreased 12-fold.

Of the children studied, 32% received at least 1 off-label drug (85.9% when unlabeled products are included). Overall, 14.7% of the pharmaceutical products administered to children were off-label and 35.8% were unlabeled and NHPs (Table 4). The most common off-label drugs were ranitidine, lansoprazole, and topical hydrocortisone (Table 3). Across all age groups, the rate of off-label use within prescription drugs use was markedly higher compared to off-label use within OTC drugs. Additionally, the rate of off-label use of OTCs was highest in infants less than 3 months old (Table 4).

Exposure to category ATC A (Alimentary tract and metabolism) pharmaceuticals was the most common among the study population (962 products/1000 children; primarily Vitamin D), followed by categories ATC N (Nervous system, 688/1000 children; primarily acetaminophen) and ATC D (Dermatologicals; 501/1000 children). 

Figure 3 shows an increase in use over time was observed for ATC codes N, M (Musculoskeletal system), J (Anti-infectives for systemic use) and R (Respiratory system). A downward trend over time in the use of ATC A drugs was also noted. There was a wide inter-center disparity in relative rates of ATC code A and J drugs use (range: 7.9–15.1%, 5.6–9.8% of drugs used, respectively). The rates of use of other groups of pharmaceutical products were similar among study sites. NHPs were used by 79.8% of the infants in the cohort.

Commonly reported indications for infant medication use by parents were: (i) discomfort from teething for the use of acetaminophen and ibuprofen, (ii) rash for topical steroids and antibiotics, (iii) gas for simethicone and gripe water, and (iv) infection for several topical, oral and ophthalmic antibiotics. Homeopathic products were administered to treat gas and discomfort from teething. Ear infection was reported as an indication for the antibiotic, amoxicillin.

Several pharmaceutical product coadministration patterns in infants were observed (Figure 4) including strong associations between use of fluticasone and salbutamol (both commonly used to treat asthma-like symptoms), and of simethicone and gripe water (both commonly used to treat colic). With the exception of lansoprazole, study site and household income differences were for the administration of OTC, including vitamin D, and NHPs. Patterns of association identified using the mutual information approach were consistent with patterns seen using Spearman rank correlation (Appendix A).

## 4. Discussion

This descriptive study documented pharmaceutical product use in 3050 Canadian infants from birth to age 12 months over 2009–2013. Almost all infants had received at least 1 pharmaceutical product, similar to or higher than the extent of medication use in older studies from the UK^4^ and Canada [16]. Prescription medications were used by 34% of infants, double the percentage reported in US children over the first 2 years of life [17]. A total of 307 different pharmaceutical products were administered to infants, of which most were available without a prescription. NHP use was frequent and accounted for 8 of the 32 most commonly used products.

Taken by 79% of infants, vitamin D was the most used pharmaceutical during infancy. This was not unexpected as Health Canada has recommended intake of 400 IU vitamin D per day for children aged 0–12 months since 2004. Prior reports on compliance with these recommendations have been restricted to infants under 2 months of age or to infants from a single city or province [18,19,20]. In our study, usage of vitamin D fell from 66% in 3-month-old infants to 50% in 7–12-month-old infants. Vitamin D administration was markedly reduced in the lowest income infants and those living in Winnipeg. These results will help target infant groups at risk for vitamin D deficiency at a time of emerging evidence on vitamin D’s protection against COVID-19 lung disease [21,22]. 

In this Canadian study, 85% of infants were given a pharmaceutical product for off-label or unlabeled indications; half of all products taken by infants were used in this context. There are several caveats to evaluating NHP product use in terms of “off-label” and “unlabeled” since these terms are not universally defined, and NHPs are notoriously difficult to classify due to country-specific variations in their regulatory approval [23]. Having said this, our rates for off-label use are of the same magnitude as other developed countries [2,24,25,26,27]. Off-label medication treatment of children is well described in the hospital setting, involving up to 65% of prescription drugs in pediatric and neonatal units [26]. Fewer studies—none in Canada—have investigated this issue from an outpatient perspective [25,26,27,28]. 

Not surprisingly, in 67 % of all infants, the second most used medication across all ages was acetaminophen [4,29,30]. This was followed by ibuprofen where usage rose from 8% of infants aged 4–6 months to 23.5% by age 7–12 months. In contrast, a large US survey showed no treatment with ibuprofen before 6 months of age [29]. The Canadian Pediatric Society recommends that ibuprofen treatment of infants be discussed with a physician [31]. According to parent reporting, the main indication for acetaminophen or ibuprofen treatment in study infants was teething discomfort. Approximately 6% of infants were given NHPs for the same purpose. The prevalence of analgesic use in our study aligns with rates of 36–50% for teething pain or fever found in other infant populations [32]. North American physicians frequently recommend analgesics or teething rings for infant teething symptoms, whereas NHPs (e.g., homeopathic drops) are more often prescribed by French pediatricians [33,34]. 

Of particular interest in our study is the use of NHPs other than vitamin D drops, seen in 1.1% of infants. Vernacchio et al. found that herbal products other than vitamins or minerals were used by fewer than 0.5% of children under age 12 years [29], based on a short 7-day assessment period. In our study, four homeopathic products were each used by at least 1% of infants, mostly for minor conditions such as colds, colic and teething. This indicates higher prevalence of homeopathy use in infants than in children as a whole, reported to be 1.3% in a large US survey [35]. A similar trend was identified in a Scottish prescription registry where homeopathic product usage was almost 8 times more frequent in infants than older children [36]. As recommended by professional associations, our findings support the need for health care provider discussion on homeopathic and NHPs with young children’s families [37]. This is especially important as less than half of Canadian parents inform their physician of their children’s use of NHPs [38].

Several trends and medication use patterns were noteworthy. Infants in our cohort were increasingly more likely to receive prescription drugs in many ATC groups with advancing age, as seen in other infant populations [1]. Alimentary tract medications represented the only ATC category to decline in use with age, which is consistent with the natural history of diminished gastroesophageal reflux and use of vitamin D supplementation by age 1 year [39]. We found that infants in our cohort had increasingly higher exposure to ATC groups H, J, M, N, and R drugs over the first year of life. This could be due to the greater comfort of both parents and physicians in giving medications to older infants. Many observed medication use associations are clinically common, for example the co-treatment of asthma with fluticasone and salbutamol. Other frequent combinations involved prescription drugs and NHPs/OTC drugs (e.g., nystatin and gentian violet), underlining the importance of providing information on OTC or NHP use when prescribing or dispensing prescriptions in infants. Apart from the proton pump inhibitor drug, lansoprazole, there were no associations between prescription medications and study city. In contrast, many study city associations were noted for OTC drugs and NHPs, likely a function of local physician preference, and the marketing and availability of OTC drugs and NHPs not covered by prescription insurance plans. 

Our study had several strengths and some limitations. Medication use in young children varies greatly in the literature [1,4], according to age and type of products queried. A US survey found that 56% of infants received at least 1 medication in the past 7 days [29], whereas in another US survey of children under age 2 years, prescription medication usage was 17% with a 30-day ascertainment period [17]. Recall bias is more common with longer periods of assessment. In this study, parents recalled medication use over a 3- to 6-month period. Except in jurisdictions where OTC drugs are reimbursed by prescription insurance plans [16], questionnaire is the only method to identify the use of a wide array of OTC and NHPs that cannot be captured in prescription database records. However, we did not specifically inquire into types of NHPs and their use may have been underreported. Finally, our study is mostly representative of infants in urban centres.

In summary, we evaluated age-specific and longitudinal use of prescription, OTC, and NHP pharmaceutical products during infancy in a large general population. Vitamin D was the most common pharmaceutical used and substantial use of off-label and NHPs was evident. This description will inform healthcare professionals in prescribing and advising on infant medication use. It will also stimulate further research into the effectiveness and safety of medication use over the first year of life.

## Figures and Tables

**Figure 1 children-09-01475-f001:**
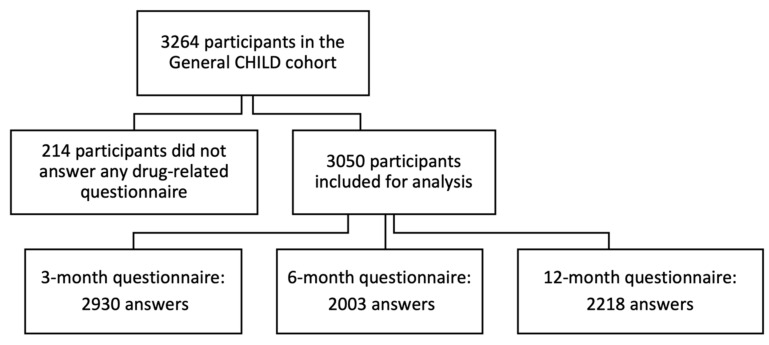
Flow chart for participants included in study.

**Figure 2 children-09-01475-f002:**
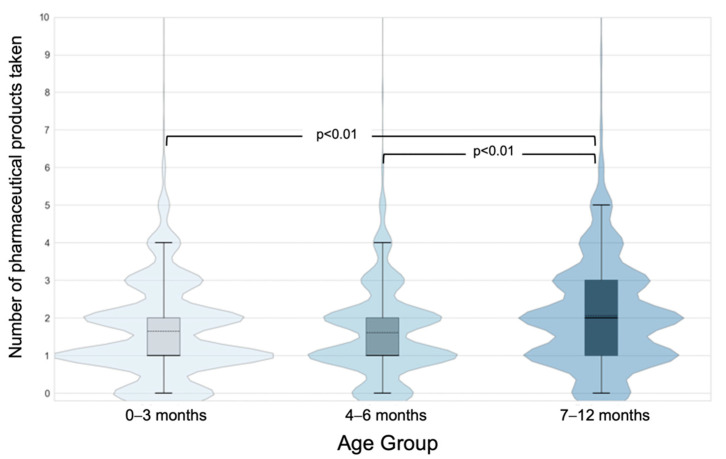
Number of pharmaceutical products (prescription, over-the-counter drugs, and natural health products) taken at each time period expressed as a violin plot and overlaid with box and whisker plot where whiskers represent the 5th and 95th percentiles. The thick and solid black line is the median and the dashed black line is the mean.

**Figure 3 children-09-01475-f003:**
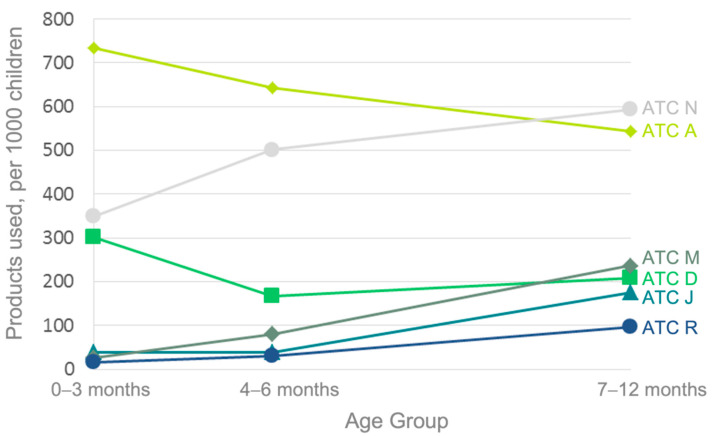
Use of drugs from the most common Anatomical Therapeutic Chemical (ATC) drug groups, by age.

**Figure 4 children-09-01475-f004:**
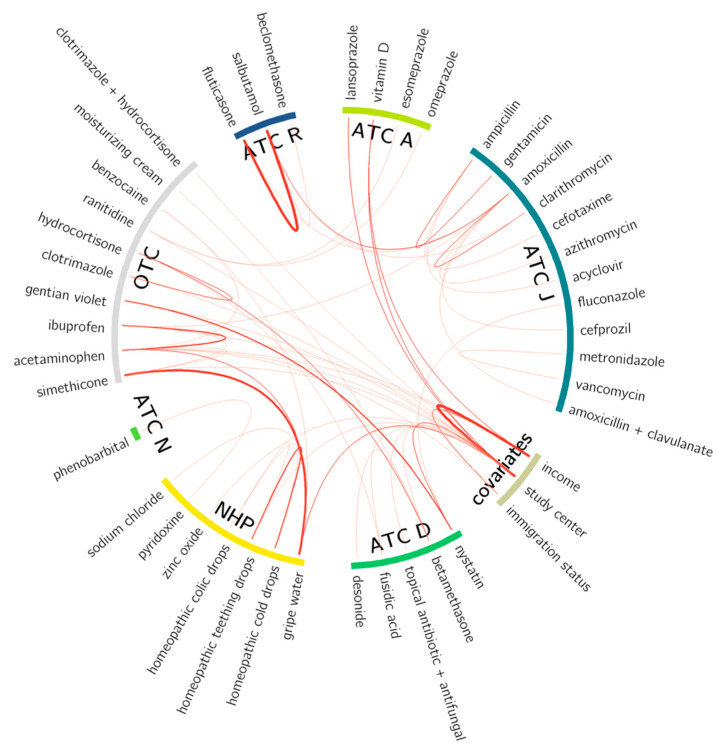
Chord diagram or “globe” showing 57 significant mutual information (MI) scores calculated for all possible pairs of medications taken together by child subjects throughout the first year of life. The thickness of the red line corresponds to MI score magnitude for pairs of medications with an MI score threshold of ≥0.005 and *p*-value corrected for multiple testing (*p* ≤ 0.05). Pharmaceutical medications are grouped into their respective top-level ATC code (alimentary tract and metabolism (A), dermatologicals (D) anti-infectives for systemic use (J), nervous system (N) and respiratory system(R)) or into natural health products (NHP) or over-the-counter (OTC) drugs.

**Table 1 children-09-01475-t001:** Characteristics of included study participants from the CHILD Cohort Study.

Characteristic	Study Participants, n (%)
**Sex**	
Female	1447 (47.4)
Male	1603 (52.6)
**Season of birth**	
Winter	753 (24.7)
Spring	816 (26.8)
Summer	763 (25.0)
Autumn	718 (23.5)
**Year of birth**	
2009	62 (2.0)
2010	779 (25.5)
2011	1669 (54.7)
2012	540 (17.7)
**Study center**	
Vancouver	679 (22.3)
Edmonton	717 (23.5)
Manitoba	922 (30.2)
Toronto	732 (24.0)
**Parental socioeconomic status**	
Low income (USD 0–29,999)	132 (4.3)
Middle income (USD 30,000–99,999)	1133 (37.1)
High income (≥ USD 100,000)	1417 (46.4)
Did not answer	368 (12.2)
**Maternal time lived in Canada**	
<5 years	216 (7.1)
≥5 years	2763 (90.6)
Did not answer	71 (2.3)

**Table 2 children-09-01475-t002:** Exposure to pharmaceutical products in the CHILD Cohort Study, by age.

Type	0–3 Months	4–6 Months	7–12 Months	0–12 Months
	(n = 2930)	(n = 2003)	(n = 2218)	(n = 3050)
Pharmaceutical products, n (%)	2449 (83.6)	1697 (84.7)	1983 (89.4)	2927 (96.0)
Prescription drugs	536 (18.3)	235 (11.7)	556 (25.1)	1036 (34.0)
OTC drugs	1391(47.5)	1150 (57.4)	1538 (69.3)	2349(77.0)
NHPs	1894 (64.6)	1213 (60.6)	1205 (54.3)	2433 (79.8)
Mean number of products, per child (SD)	1.65 (1.28)	1.61 (1.21)	2.07 (1.48)	2.99 (1.93)
Prescription drugs	0.24 (0.58)	0.16 (0.51)	0.38 (0.82)	0.56 (1.00)
OTC drugs	0.60 (0.74)	0.73 (0.76)	1.00 (0.84)	1.28 (1.03)
NHPs	0.77 (0.7)	0.69 (0.65)	0.63 (0.68)	1.07 (0.83)

Note: OTC = over the counter, NHPs = natural health products, SD = standard deviation.

**Table 3 children-09-01475-t003:** Prevalence of use of most used pharmaceutical products, by age.

Product	ATC Code	Type	Prevalence of Use (%)				Most Common Reported Reasons for Use
			0–3 m	4–6 m	7–12 m	0–12 m	
			(n = 2930)	(n = 2003)	(n = 2218)	(n = 3050)	
Vitamin D	A (A11CC)	NHP	66.2	56.9	50.1	78.7	Supplement
Acetaminophen	N (N02BE01)	OTC	33.9	49.7	58.7	67.3	Discomfort, teething
Ibuprofen	M (M01AE01)	OTC	2.6	8	23.5	20.4	Discomfort, teething
Topical hydrocortisone	D (D07XA01)	OTC	5.6	5.2	6.9	11.3	Eczema, rash
Amoxicillin	J (J01CA04)	Rx	1.2	2.2	11	9.7	Ear infection
Nystatin	D (D01AA01)	Rx	8.2	1.8	2.2	9.7	Candidiasis
Simethicone	A (A03AX13)	OTC	7.6	2.7	0.6	8.4	Gas
Gripe water	N/A	NHP	6.3	2.7	0.9	7.3	Gas
Topical zinc oxide	D (D02AB)	NHP	4.6	1.6	2.2	6.3	Rash, diaper dermatitis
Topical clotrimazole	D (D01AC01)	OTC	3.2	0.6	1.9	4.6	Rash, diaper dermatitis
Homeopathic teething drops	N/A	NHP	0.7	2.9	3.5	4.5	Discomfort, teething
Ranitidine	A (A02BA02)	OTC	3.7	2.1	0.9	4.2	Gastroesophageal reflux
Salbutamol	R (R03AC02)	Rx	0.9	1.3	3.2	3.5	Wheezing
Sodium chloride	N/A	OTC	1.9	1	1.4	3.2	Nasal congestion
Homeopathic cold drops	N/A	NHP	1	1.1	1.9	2.7	Common cold
Diphenhydramine	D (D04AA32)	OTC	0.1	0.6	2.6	2.3	Allergy
Topical benzocaine	N (N01BA05)	OTC	0.3	1.8	1.3	2.1	Discomfort, teething
Topical fusidic acid	D (D06AX01)	Rx	1	0.5	0.9	2	Rash
Topical erythromycin	S (S01AA17)	Rx	1.1	0.3	0.5	1.6	Eye infection
Lansoprazole	A (A02BC03)	Rx	1	0.9	0.7	1.5	Gastroesophageal reflux
Dexamethasone	C (C05AA09)	Rx	0.2	0.2	1.7	1.5	Croup
Topical polymyxin B + gramicidin	S (S01AA30)	OTC	0.8	0.6	0.6	1.5	Eye infection
Clarithromycin	J (J01FA09)	Rx	0.1	0.2	1.7	1.4	Bronchitis
Homeopathic teething tablets	N/A	NHP	0.2	1	0.9	1.4	Teething
Probiotics	A *	NHP	1	0.6	0.3	1.4	Intestinal flora
Fluticasone	R (R01AD08)	Rx	0.2	0.5	1.4	1.3	Wheezing
Gentian violet	D (D01046)	OTC	1.4	0.2	0	1.3	Candidiasis
Moisturizing creams	D *	OTC	0.5	0.8	0.4	1.2	Dry skin
Cephalexin	J (J01DB01)	Rx	0.3	0.3	0.8	1.1	Urinary tract infection, skin infection
Azithromycin	J (J01FA10)	Rx	0.1	0.2	1.2	1	Ear infection, chest infection
Topical betamethasone	D (D07AC01)	Rx	0.3	0.3	0.7	1	Eczema, rash
Homeopathic colic drops	N/A	NHP	0.9	0.3	0.1	1	Gas, colic

Abbreviations used: m = months, ATC = Anatomical Therapeutic Chemical, OTC = over the counter, NHPs = natural health products, Rx = prescription medication, N/A = Not applicable. * No detailed ATC code available for this pharmaceutical product.

**Table 4 children-09-01475-t004:** Proportion of off-label and unlabeled pharmaceutical products, by age.

Type	0–3 Months	4–6 Months	7–12 Months	0–12 Months
Pharmaceutical products used, n	4825	3221	4587	9127
Off-label drugs, n (%)	535 (11.2)	317 (9.9)	471 (10.4)	1342(14.7)
NHPs and unlabeled products	2267 (47.0)	1383 (43.0)	1404 (30.6)	3267(35.8)
Prescription drugs used, n	696	319	836	1705
Off-label prescription drugs, n (%)	189 (27.2)	142 (44.5)	233 (27.9)	493 (28.9)
OTC drugs used, n	1769	1469	2215	3895
Off-label OTC drugs, n (%)	346 (19.6)	175 (11.9)	238 (10.7)	849 (21.8)
Medication Unknown, n	93	50	132	260

Note: Total pharmaceutical products taken (top row) is the sum of NHPs and unlabeled products taken, Prescription drugs taken, OTC drugs used, and medications reported as “unknown” in the questionnaire. Unknown medications have no further assignments as prescription, OTC or NHP pharmaceutical products. Abbreviations used: OTC = over the counter, NHPs = natural health products.

## Data Availability

Data is available upon request from https://childstudy.ca/for-researchers/child-db/ (accessed on 7 October 2021).

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
