# Peer review of "From Prescription Drugs to Natural Health Products: Medication Use in Canadian Infants"

_children, 2022, doi:10.3390/children9101475_

Round 1
Reviewer 1 Report
Thanks for this interesting work. Please find enclosed my comments and suggestions to bring more light to some aspects of the manuscript. Some results and calculations would need revision for the sake of clarity and fluency when reading the text. I would also suggest checking all calculations and formulas used since many errors have been identified. Note that some comments affect several sections of the manuscript at once, also the abstract.
Abstract.
- Why vitamins were assessed separately? Which is their regulatory status in Canada (ie, food supplement or drug)? Please provide clarification on that aspect (also in Methodology section), as it seems that they are an independent product category.
- Homeopathic products appear here but after no data is disclosed in the manuscript until the Discussion section. Please add the data if any, clarify the status or otherwise eliminate any reference to that products in the Abstract.
- In the following sentence “Vitamin D, acetaminophen, ibuprofen, topical hydrocortisone, amoxicillin, and nystatin were the most common medications and natural health products (NHPs) received”, there is no NHP in the list. Please rephrase to clarify.
In addition to that, it would be helpful to include how much these 6 products accounted for (%).
- According to the following sentence “Overall, 14.7% of pharmaceutical products administered to children were off-label and 35.8% were NHPs or products without a Drug Identification Number (DIN)”, how could you assess if they were off-label or not since they do not have a product monograph or PIL? In the Methodology section I understood that only drugs could be assessed, so please clarify here.
- % of off-label OTC is missing in the abstract.
Introduction.
- According to the sentence “Very few data exist regarding medication use in young children” I found that in Europe there have been some studies on the topic, and also about off-label use (see examples below that you can use, as they are more recent than some of the referenced in the manuscript).
- https://doi.org/10.3390/pharmaceutics13040588
- https://doi.org/10.1002/pds.5354
- doi: 10.1111/jebm.12402
- PMID: 31379392
- https://ec.europa.eu/health/sites/health/files/files/documents/2017_02_28_final_study_report_on_off-label_use_.pdf
- I would suggest focusing the introductory idea in Canada, what the background is and which are the unmet needs or gaps. For instance, is there any national initiative that encourages pediatric drug research? Any local regulation? As the EMA one, for instance: Paediatric Regulation (1901/2006/EC).
Materials and Methods.
- Please indicate which years comprise the CHILD study.
- In case of the patient being in the hospital, was the data about medication use collected? In addition to that, it would be useful to add here some kind of inclusion/exclusion criteria on the data to be reported in the questionnaires.
- Page 2 Line 64. Could you clarify the aim of the sentence “Newborns with factors that could confound studies on development of wheezing were excluded”. Since I see in Table 3 that drugs used for wheezing were included.
- Page 2 Line 72. What do you mean by “similar questionnaires”? Could you develop more the idea here? Which additional questions were raised? Was it possible to determine the aim for each product used?
- Page 2 Line 75. In the sentence “When recording any medication usage between 0-12 months, missing survey responses (unreturned questionnaires) were permitted. When recording absence of medication usage over the same period, missing responses were prohibited” I cannot understand the rational.
- In the sentence “Products were attributed an Anatomical Therapeutic Chemical (ATC) code, using the ATC Index 2015” more clarification would be needed, as ATC is exclusively intended for active substances (drugs). NHP may not be classified with ATC so please specify that and find another classification system in other comparable studies if needed. In case of homeopathy, how was the classification managed?
- Please define “foreign products”. Are they imported drugs or are they drugs that patients directly buy in other countries?
- Again here, Vitamin D appears and it seems it is a nutrition product. Are vitamins (and their combinations) the only nutrition products assessed in the study?
- Page 2 Lines 89-95. Do NHP have any kind of patient information leaflet (PIL) that you could check for off-label use? Please have a look here: https://www.canada.ca/en/health-canada/services/drugs-health-products/natural-non-prescription/applications-submissions/product-licensing/licensed-natural-health-products-database.html
- According to the sentence “Descriptive statistics were used to depict use of pharmaceutical products in different age periods”. Please indicate which were the age periods analysed.
- In this section, a more detailed description of all the variables and covariates collected, the data analysed and the formulas used to get each result would be needed for the sake of clarity. The latter (formulas) could be added as a Suppl Material.
- There is no clue for homeopathic products. How were they assessed?
Results.
- Given the extension of this section, it would be recommended to add some subheadings, which will help structuring the text. Replicate them for the Discussion section as well.
- Page 3 Line 109. 3050 is 84.2% of 3624, not 93%.
- Page 3 Lines 111-12. Likewise, percentages do not match (2930/3050=96% and 2003/3050=65.7%). Please refer to my previous comment in Methodology section: a detailed formula for each calculation will be extremely useful.
- Figure 1. There is a mismatch in total numbers, as 214+3050 is not 3624.
- Table 1. Why season of birth is important? Please refer to my previous comment in Methodology section: description of all variables collected.
- In addition, the sum of all rates is 100.1%, which needs to be amended. From my calculations I would say that autumn is 23.5% (718/3050).
- On the other hand the year of birth rates accounts for 99.9%, which also needs revision. Parental socioeconomic status: 0.1% needs to be added in “did not answer category”.
- In case of “maternal time lived in Canada”, rates are not correct and would need to be amended.
- Statistics in Tables 1, 2 and 3. Are there any significant statistical differences between the categories of each variable? Please add that information to the tables (and indicate the analysis carried out in the Methodology section as well).
- Page 4 Line 134. NHP and homeopathic ones are not the same. I would suggest reporting the results separately. Review carefully the homeopathic products presence along the manuscript, as it is not clear and leads to confusion.
- Table 2 and Figure 2. Where are vitamins and homeopathy included?
- Table 3. In case of drugs, I would suggest adding more detail in the ATC information, not only the category (A, B, C…) but also therapeutic subgroup for instance.
- Page 4 Lines 133-35. Just to size the importance of these 16 products, I would suggest adding also the percentages for each category (system Ab, topical anti-infectives, NHP). In addition, there are 7 NHP but in the abstract these ratio is depicted as 8/32. Please confirm and amend.
- After this sentence, it is confusing reading that Vitamin D is one of the most used products, as it is not a system Ab, topical anti-infective or NHP. I would suggest rephrasing the paragraph.
- Page 4 Lines 135-37 and Table 3. I see that it was possible to determine in the questionnaires the aim for each product used. As stated before, it is not explained in the Methodology section and would be a must.
- Table 3. Here Vitamin D appears classified as a drug. Please review the text and clarify it the product is a drug or NHP in Canada, since certain sentences may lead to confusion. Also, why OTC drugs are not classified with ATC code? They should.
- Table 4. Totals do not match with the sum of colored lines, ie 2267+696+1769 is not 4925. Review and amend.
- Page 5 Line 142-48 and Table 4. I would suggest checking the labelling of NHP in the canadian database shared (https://www.canada.ca/en/health-canada/services/drugs-health-products/natural-non-prescription/applications-submissions/product-licensing/licensed-natural-health-products-database.html) and then reorganize the results of NHP use off-label or not.
- According to the 3 reasons given to consider off-label use (Methodology section), which has been the ranking between them?
- Figure 3. Where the differences between groups statistically significant? (ie, for each category and between the 3 age groups).
- Page 6 Lines 171-77. It would be needed to develop more the idea of the covariates and the differences in results that have been detected.
- In addition to that, if the differences are statistically significant or not (ie, do high income status consume significantly more homeopathic products? “” consume significantly more off-label? etc etc).
Discussion. Consider all the amendments suggested in the Results section to drive the Discussion.
- Page 7 Line 191. “32 different pharmaceutical products were administered to infants”. It is not completely correct, I understood the following from Results section: “32 different pharmaceutical products were administered to at least 1% of infants” and that “307 different pharmaceutical products were administered to infants”.
- The sentence “NHP use was frequent and accounted for 7 of the 32 products”. In the abstract these ratio is depicted as 8/32. Please confirm and amend.
- Page 7 Line 200. What does “late infancy” mean here?
- Page 7 Line 196. Was this recommendation of Vitamin D use in force during the study period (2009-13)? If not it would not be useful for the discussion.
- Page 7 Line 201. Is there any explanation for that situation? “Vitamin D administration was markedly reduced in the lowest income infants and those living in Winnipeg”.
- Page 8 Line 203. I do not see the link with covid-19 here, as the study data is until 2013. Any current recommendation would not apply to the study data or medication use patterns.
- Page 8 Line 204. Not sure if this sentence is correct “half of infants were given a pharmaceutical product for off-label or unlabeled indications; 30% of all products taken by infants were used in this context”. Results section gives different perspective: 32% of children and 50.5% of medicinal products + NHPs (see Page 5 Lines 142-48). Please check and rephrase if needed.
- Page 8 Line 210. There are more recent studies/references that could be assessed for inclusion here.
- https://doi.org/10.3390/pharmaceutics13040588
- https://doi.org/10.1002/pds.5354
- Page 8 Line 210. There is a reference needed when talking about the hospital setting.
- Figure 3 (Results section). Discuss more in deep why some categories increase and others decrease during the 12-month development of the infant.
- Figure 4. Discuss more in deep the most relevant results in the diagram.
- Page 9 Line 263-68. Not only prescription drugs and OTC were assessed.
- Also, some conclusions about off-label use and covariates should be included to close the manuscript.
Supplementary materials.
- Page 9 Line 271. 42 medications are disclosed but the study is talking about 32 all the time. Please check the content and calculations of this file.
Author contributions.
- There are some authors for whom their contribution is not stated.
Author Response
We thank all reviewers for taking the time to provide valuable feedback that strengthens the overall quality of our paper. We’ve considered all comments and addressed them point-by-point below in red font in the attached PDF. In response to suggestions, we’ve incorporated changes to the manuscript and corrected typographical errors where applicable.
Reviewer 2 Report
Dear Authors,
Thank you for this interesting manuscript. Data regarding medications used by infants (especially for outpatient setting) is scarce. Thus, data provided in this manuscript is valuable and relevant for clinical practice. To the best of my knowledge, methodology and statistical analysis are adequate, results are clearly and nicely presented, potential limitations are noted, and conclusions are supported by the results.
Hence, I have only minor suggestions to add:
(I) Do you have separate data (or can you perform and enclose it) regarding prescription rates among term vs preterm infants? I believe it would be interesting and useful to overview it.
(II) Do you have info how many pharmaceutical products were prescribed/recommended by their physician and what was the percentage of pharmaceutical products taken by parents’ own discretion or given by accident (in general, per individual periods and per individual product groups)? If yes, please present.
(III) Reference list can be updated with newer publications.
Best regards, Reviewer
Author Response
We thank all reviewers for taking the time to provide valuable feedback that strengthens the overall quality of our paper. We’ve considered all comments and addressed them point-by-point in red font in the attached PDF. In response to suggestions, we’ve incorporated changes to the manuscript and corrected typographical errors where applicable.

Round 2
Reviewer 1 Report
Many thanks for all the clarifications provided, specially in terms of NHPs, vitamins and homeopathic products , as well as its analysis throughout the manuscript.
I have reviewed in detail all the responses provided and still have some minor comments to disclose. Please find them below.
Abstract
- Still vitamins are separated from NHP in the following sentence: “Parents were surveyed for prescription and over-the-counter drugs, natural health products (NHPs), homeopathic products and vitamin use at 3, 6, and 12 months after delivery.”
Methodology
- Page 2 Lines 64-74. Same sentences are repeated twice.
- Still need some kind of complete list or description of the variables and covariates collected. Most are explained in the section but other directly appear in the Results section (ie season of birth etc).
Results
- Table 3. I would strongly encourage adding more detail on the ATC code reported for each product. I completely understand that overcrowding the table is not useful but, this would be the key for comparison with other studies. For instance:
Vitamin D - instead of putting “ATC A” just put “A11CC” (same lenght in number of characters and more explanatory data). Info available here: https://www.whocc.no/atc_ddd_index/
- Page 5 Line 145. Typo - A space is needed here “and4”.
- Table 4. I would recommend adding a footnote with the response you provided to my comment, which explains why totals do not match. It would help improving the clarity of the data in the table.
Discussion
- Page 8 Line 206. “32 different pharmaceutical products were administered to infants”. It is not completely correct, I understood the following from Results section: “32 different pharmaceutical products were administered to at least 1% of infants” and that “307 different pharmaceutical products were administered to infants”.
- According to the “late infancy” phrase, it would be better understood as “7-12 months” from my point of view. Otherwise, if you defined previously what “late infancy” meant (ie, in the latest paragraph of the Methodology section) you can put “late infancy” on the Discussion.
Author Response
We thank you for taking the time to provide a second round of feedback that strengthens the overall quality of our paper. We’ve considered all comments and addressed them point-by-point below in red font below. In response to suggestions, we’ve incorporated changes to the manuscript, corrected typographical errors where applicable, and revisions recorded using “track changes”
